# Molecular Structure, Vibrational Spectrum and Conformational Properties of 4-(4-Tritylphenoxy)phthalonitrile-Precursor for Synthesis of Phthalocyanines with Bulky Substituent

**DOI:** 10.3390/ijms232213922

**Published:** 2022-11-11

**Authors:** Natalia V. Tverdova, Nina I. Giricheva, Vladimir E. Maizlish, Nikolay E. Galanin, Georgiy V. Girichev

**Affiliations:** 1Department of Physics, Ivanovo State University of Chemistry and Technology, Sheremetevsky Avenue 7, 153000 Ivanovo, Russia; 2Nanomaterial Research Institute, Ivanovo State University, Ermak Street 39, 153025 Ivanovo, Russia; 3Department of Fine Organic Chemistry and Technology, Ivanovo State University of Chemistry and Technology, Sheremetevsky Avenue 7, 153000 Ivanovo, Russia

**Keywords:** gas electron diffraction, mass spectrometry, DFT calculations, IR spectrum, molecular structure, 4-(4-tritylphenoxy)phthalonitrile, conformers

## Abstract

By DFT method with B3LYP, PBE, CAM-B3LYP, and B97D functionals, it was found that the molecule 4-(4-tritylphenoxy)phthalonitrile (TPPN) has four conformers. The geometric structure, vibrational frequencies, electronic characteristics, and thermodynamic functions of conformers, as well as the structure and energy of transition states, were determined. IR spectrum of TPPN film contains vibrational bands belonging to different conformers. The assignment of bands was performed basing the distribution of normal vibration energy on internal coordinates. A synchronous electron diffraction/mass spectrometric experiment was performed to determine the structure of conformers in a saturated TPPN vapor. The elemental composition of the ions recorded in the mass spectrum indicates the thermal stability of TPPN at least up to T = 200 °C. The difference in the structure of tetrasubstituted metal phthalocyanines, which can be synthesized from different TPPN conformers, has been shown.

## 1. Introduction

Currently, considerable attention is paid to phthalocyanines containing bulk trityl, due to their manifestation of liquid crystal properties [1,2,3,4], as well as the prospects for their use in thin-film electronics [5]. In addition, phthalocyanine compounds with high solubility are currently needed [6,7]. An increase in solubility can be achieved by introducing substituents of a certain nature [7,8]. In [5], novel tetra (4-tritylphenoxy) substituted copper, zinc, and cobalt phthalocyanines were obtained by reaction of 4-(4-tritylphenoxy)phthalonitrile (TPPN) (Figure 1) with CuCl_2_, ZnCl_2_, CoCl_2_ metal salts (scheme in two dimensions of metal substituted phthalocyanine molecules is given in Appendix A). It has been shown in Ref. [5] that the presence of bulky 4-tritylphenoxy substituents on the peripheral positions prevents the aggregation in organic solvents, such as CHCl_3_, CH_2_Cl_2_, THF, DMF, DMSO, and adds interesting properties to phthalocyanines, such as high solubility.

It is well known that tetrasubstituted phthalocyanines are usually obtained as a mixture of four constitutional isomers [9]. This can be considered a positive or negative factor, depending on the purpose of applied phthalocyanines. For using these compounds as materials in thin films, it is even better to have an isomeric mixture that will not crystallize very easily [2]. The presence of bulky 4-tritylphenoxy substituents in the β-position can lead to regioselectivity of substituted phthalocyanines.

However, 4-tritylphenoxy substituents have a structural feature that is usually overlooked and which can affect the thermodynamic, spectral, and electro-optical properties of metal phthalocyanines. This feature of 4-tritylphenoxy substituents is related to the conformational properties of TPPN, which is used as the precursor for the synthesis of phthalocyanines with trityl groups as peripheral substituents.

Obtaining reliable experimental data for the 4-(4-tritylphenoxy)phthalonitrile and their subsequent study is an actual problem due to the lack of any experimental work on the structure of this compound.

This paper describes the first experimental study (gas electron diffraction in combination with mass spectrometry, as well as IR spectroscopy) of the molecular structure and conformational properties of 4-(4-tritylphenoxy)phthalonitrile supplemented by quantum chemical calculations.

## 2. Results and Discussion

### 2.1. DFT Calculations

#### 2.1.1. Conformational Analysis

The TPPN molecule has six torsion coordinates, which can give rise to several conformers that differ in the mutual orientation of trityl, phenoxy (PhO), and phthalonitrile (PhN) fragments. The atom numbering is shown in Figure 1.

To determine the structure of the found conformers (Figure 2) and the barriers between them, the potential functions of internal rotation were calculated (B3LYP/cc-pVTZ): Around the C_O1_-O (Figure 3), C_t_-C_O4_ (Figure 4), C_N1_-O (Figure 5) bonds, and around C_t_-C_t1_ bond in the trityl group. The ϕ[C_O2_C_O1_OC_N1_], ϕ[C_O3_C_O4_C_t_C_t1_], and ϕ[C_O1_OC_N1_C_N2_] dihedral angles were scanned with a step of 2° or 10°.

The potential function of the internal rotation of the O-PhN group around the C_O1_-O bond has two deep minima (Figure 3) corresponding to *cis*-conformers I and II with ϕ[C_O1_OC_N1_ C_N2_] ≈ 0° (Table 1). The potential function of the internal rotation of the O-PhN group for *trans*-conformers III→IV with torsion angle ϕ[C_O1_OC_N1_C_N2_] ≈ 180° (Table 1) has a similar form. In *cis*-conformers, the C_O1_–O bond eclipses the C_N1_-C_N2_ bond, whereas in *trans*-conformers, the dihedral angle C_O1_OC_N1_C_N2_ is approximately 180°. As a result, *cis*-conformers I and II differ from the corresponding *trans*-conformers III and IV by the position of the C≡N substituents in the phthalonitrile moiety (Figure 2).

The barrier between *cis*-conformers I and II and, accordingly, between *trans*-conformers III and IV is practically the same and amounts to 6.3 kcal/mol. In transition states TS_I→II_ and TS_III→IV_ the dihedral angle ϕ[C_O2_C_O1_OC_N1_] between the planes of the PhO and PhN fragments is close to 0° (Figure 2 and Figure 3).

The second way of the transition between conformers I→II and III→IV can be realized by rotating the PhO-PhN fragment around the C_O4_-C_t_ bond. In this case, in the TS_I→II_ and TS_III→IV_ transition states, strong steric interaction occurs between PhO and trityl C(Ph)_3_ fragments, which leads to a high energy barrier > 16 kcal/mol (Figure 4).

The difference in the two *cis*-conformers I and II, as well as in the two *trans*-conformers III and IV, lies in the different orientation of the PhN group relative to trityl fragment, namely, in conformers I and III, the dihedral angle C_O2_C_O1_OC_N1_ should be defined as φ = +99°, and in conformers II and IV φ = −95° (Figure 3, Table 1).

Figure 5 demonstrates that *cis*→*trans* transition can be realized by rotation of the PhN fragment around the O-C_N1_ bond. The energy barrier between the *cis*-conformer II and *trans*-conformer IV is 5.6 kcal/mol and corresponds to TS_II→IV_ in which the bond angle C_O1_-O-C_N1_ between the PhO and PhN fragments is 118°, and dihedral angle ϕ[C_O1_OC_N1_C_N2_] is 90°.

Additional calculations were carried out to estimate the barrier of internal rotation of the phenyl ring of the trityl group around the C_t_-C_t1_ bond. It has been shown that an independent rotation by ~40° of one of the phenyl fragments leads to a strong steric repulsion between the Ph fragments in trityl. As a result, the phenyl rings cannot rotate but perform synchronous torsional vibrations. The amplitude of these vibrations Δϕ depends on the thermal energy RT and is estimated as ±16° at T = 298 K and ±18° at T = 478 K.

#### 2.1.2. The General Structural Motives and Electronic Characteristics of Conformers

Theoretical calculations performed using four DFT functionals lead to consistent results. All variants of the DFT method predict the presence of four C_1_ symmetry conformers for the TPPN molecule. For identical molecular forms, the differences in the corresponding internuclear distances do not exceed 0.002 Å, in the values of bond angles 1°, and less than 7° in the values of torsion angles (Appendix A).

All conformers have a similar structure of (Ph)_3_C(PhO) fragment. The central site C(C)_4_ is a distorted tetrahedron with identical C-C bonds, two C-C-C bond angles that are smaller than the tetrahedral one, and four C-C-C bond angles larger than the tetrahedral one (Appendix A). The bond lengths and bond angles in this fragment are the same in all conformers. Differences are observed in the values of torsion angles of *cis*- and *trans*-conformers as well as bond lengths in the PhN fragment. In addition, in conformers I and III, the shortest distance H···H between the hydrogen atoms of the trityl group and the phthalonitrile PhN fragment is ~3.7 Å and in conformers II and IV ~4.1 Å.

Table 1 shows the selected characteristics of conformers of 4-(4-tritylphenoxy)phthalonitrile molecule according to DFT/B3LYP/cc-pVTZ (Appendix A shows the characteristics of the conformers calculated with the functionals B97D, CAM-B3LYP, PBE).

All conformers have close energies of frontier orbitals and large dipole moments (Table 1), however, the direction of dipole moments in *cis*-conformers differs significantly from their direction in *trans*-conformers.

Figure 6 shows the frontier MOs of conformer II. At the electron transition from HOMO to LUMO, the electron density is transferred from the trityl fragment to phthalonitrile PhN. The frontier orbitals of other conformers have a similar form.

Table 1 lists the relative energies of conformers ∆E_total_, relative Gibbs free energies ΔG°_T_, and mole fractions χ_i_ of conformers in saturated vapor calculated for T = 298 K and T = 478 K (the latter corresponds to the temperature of the GED experiment). The difference in the ratio of ∆E_total_ and ΔG°_T_ for the four conformers is due to the entropy factor.

The conformers have close electronic energies. The difference in total energy ΔE between conformers I-IV does not exceed 0.4 kcal/mol. This increases the probability that the 4-(4-tritylphenoxy)phthalonitrile may contain different conformations in the unit crystal cell and have a complex conformational composition of the gas phase (Table 1 and Appendix A). Since the barriers between conformers are quite high and exceed 5 kcal/mol, conformational transitions are unlikely even at the temperature of GED experiment (the thermal energy RT is less than 1 kcal/mol). Thus, conformational diversity may arise during the synthesis of a compound.

In this case, the mixture of conformers should be characteristic of both the condensed and gas phases. We tried to find the validity of this assumption when interpreting the IR spectra and GED data.

### 2.2. Vibrational Spectrum

The infrared spectrum for the films of TPPN is shown in Figure 7, along with the calculated spectrum for a mixture of *cis-* and *trans*-conformers.

The harmonic vibrational frequencies were calculated for the fully optimized geometries of I–IV conformers at B3LYP/cc-pVTZ level. Analysis of the calculated IR spectra showed that the spectra of the two conformers I and II (*cis*) and two conformers III and IV (*trans*) almost coincide (Appendix A).

In Table 2, the experimental vibrational frequencies ν_exp_ of TPPN recorded in this work and in [5] are compared with the calculated frequencies ω_theor_, which have the highest intensity in the corresponding spectral region for *cis*- and *trans*-conformers (Figure 7).

The values of the experimental frequencies ν_exp_ in the two works are in good agreement with each other. However, in Ref. [5], low-intensity bands at 1419, 1311, and 1033 cm^−1^ were not noted, and an erroneous assignment of the frequency at 1253 cm^−1^ to the C-O-C bending vibration was given.

The correctness of the assignment of frequencies was checked on the basis of the dependencies ω_theor_ = f(ν_exp_), which had a linear character with a correlation coefficient almost equal to 1 (Figure 8).

For all frequencies of normal vibrations, the distribution of potential energy (DPE) over internal vibrational coordinates is calculated. For this, the VibModule [10] program was used. The last column of Table 2 shows the natural coordinates that make the maximum contributions to an individual normal vibration.

The relative intensities are indicated for the vibrational bands. The assignment for all 168 vibrational frequencies of the four conformers of the TPPN molecule is given in Appendix A.

It should be noted that most bands in the IR spectrum possess a mixed nature (Table 2). For this reason, it is possible to discuss only the predominant contribution of various internal coordinates to the normal vibration. The exceptions are the stretching vibrations of C-H bonds, which are completely separated from other vibrations, and the stretching vibrations of C≡N bonds.

The bands in the range 1593–1491 cm^−1^ correspond to the vibrations of ν(C-C)_PhN_, δ(CCC)_PhN_, δ(CCH)_PhN_ in the phthalonitrile fragment. The bands at 1282, 1250, 1213, and 1176 cm^−1^ can be assigned to ν(C-O)_PhN,_ ν(C-O)_Ph_ stretching vibrations in combination with ν(C-C), δ(CCH) modes. Note that in [1,5], the band at 1250 cm^−1^ is assigned to the Ph-O-Ph bending vibration instead of the C-O stretching vibration (although the force constant stretching of C-O bond (5.38 mdn/Å) is significantly higher than the force constant deformation of the C-O-C angle (1.20 mdn/Å)).

The frequencies associated with bending vibrations of the central site C(C)_4_, phthalonitrile PhN, phenoxy PhO fragments, and the trityl group C(Ph)_3_, according to calculations, are in the range below 1000 cm^−1^. The low frequency region of the IR spectrum shows the presence of bands at 835, 752, 704, 632, 522 cm^−1^ associated with out-of-plane vibrations of the C-H, C_PhN_-C_N_ bonds.

The main differences between the calculated IR spectra of the *cis-* and *trans*-conformers are in the intensity of the band at 1566 cm^−1^ as well as the presence of characteristic bands at 1311 cm^−1^, 1442 cm^−1^ belonging to *cis*-conformers, and 1033 cm^−1^, 1419 cm^−1^ belonging to *trans*-conformers (Table 2).

The bands at 1442 cm^−1^ (*cis*) and 1419 cm^−1^ (*trans*) correspond mainly to the stretching vibrations of the C-C bonds in the phthalonitrile fragment. It should be noted that the r(C_N3_-C_N4_)_PhN_ bond length in *cis*-conformers is shorter than in *trans*-conformers (Table 3), which is reflected in the difference in the ν(C-C)_PhN_ frequency.

An analysis of the recorded vibrational frequencies allows us to conclude that the IR spectrum indicates the presence of the *cis*- and *trans*-conformers of TPPN in the condensed state.

### 2.3. GED Structural Analysis

Geometry of each conformer of the TPPN molecule possessed C_1_ symmetry and was described by 79 independent parameters. To reduce the correlation, the number of independent parameters was restricted to 14: C_t_-C_O4_, C_O4_-C_O3_, O-C_O1_, C_N3_-C, C-N, C_t2_-H bond distances, C_O4_C_t_C_t1_, C_O3_C_O4_C_t_, C_O2_C_O3_C_O4_, C_N1_OC_O1_, C_t3_C_t2_H valence angles, and C_O3_C_O4_C_t_C_t1_, C_N1_OC_O1_C_O2_, and C_t1_C_t_C_O4_C_t1′_ dihedral angles (Figure 1). The remaining 65 independent parameters were linked to the above-mentioned 14 parameters by the difference between non-equivalent structural parameters of the same type obtained from B3LYP/cc-pVTZ calculations.

VibModule program [10] was applied to calculate the vibrational corrections Δ*r* = *r_h_*_1_ − *r_a_* and the starting values of root-mean-square vibrational amplitudes of each conformer at the temperature of the GED experiment using the harmonic approximation and also taking into account the non-linear interrelation between internal and Cartesian vibrational coordinates. Starting values of the geometrical parameters and vibrational amplitudes were taken from B3LYP/cc-pVTZ calculations. The dependent parameters were determined in terms of the geometrically consistent *r_h1_*-structure. The amplitudes were refined in groups corresponding to the different peaks on the radial distribution curve. The analysis of the electron diffraction intensities was carried out using the modified KCED-35 program [11].

Theory predicts that despite the different mutual orientations of the phthalonitrile and trithyl groups, the corresponding bond lengths and most nonbonded distances of conformers I–IV are very close. This indicates that the electron diffraction intensities may be insensitive to the conformational composition of this compound.

The calculated Gibbs free energies at the temperature of the GED experiment 478 K indicate that the ratio between the mole fractions of the *cis*- and *trans*-conformers in the gas phase is close to 1 (Table 1). In addition, DFT calculations predict that the *cis* I and *trans* IV forms predominate in vapor over the *cis* II and *tarns* III forms.

To assess the ability of the GED method to identify TPPN conformers I-IV, the theoretical functions sM(s) were calculated separately for each conformer based on structural parameters (bond lengths, angles, vibrational amplitudes, and shrinkage corrections) obtained using B3LYP/cc-pVTZ level (Appendix A), as well as theoretical radial distribution functions f(r) (Appendix A). Only a small difference in the f(r) functions is observed in the range of 3–8 Å for the *cis*- and *trans*-conformers.

The least-squares analysis of GED experimental intensities was performed under the assumption that the vapor contains separate conformers, as well as under the assumption that the vapor contains *cis*- and *trans*-conformers in mole percent of 30/70, 50/50, and 70/30. In these versions of the least squares analysis, the value of the disagreement factor R_f_ varied in the range of 4.13–4.28%. In accordance with the Hamilton criterion at 0.05 significance level [12], GED data do not allow giving preference to any variant of the vapor composition.

Despite the lack of success in the experimental solution of the conformational problem of TPPN, the GED method still allows us to obtain reliable structural parameters of the molecule. Table 3 shows that the calculated geometric parameters of equilibrium configurations of *cis-* and *trans*-conformers agree satisfactorily with the experimental ones.

The comparison of the experimental functions sM(s) and f(r) with the theoretical curves for the mole percent of 50/50 *cis*/*trans* vapor model are shown in Figure 9 and Figure 10, respectively.

The performed analysis of IR spectra, data, and the results of quantum chemical calculations indicates the existence of *cis*- and *trans*-conformers of TPPN, both in the condensed and in the gaseous state, and GED/MS data do not contradict this conclusion.

Figure 11 shows examples of positions of 4-tritylphenoxy substituents that can occur in tetrasubstituted metal phthalocyanines (MPc) during their synthesis by reaction of *cis*- (a) or *trans*-conformers (b) 4-(4-tritylphenoxy)phthalonitrile with metal salts. There are significant differences in the structure of the MPc conformers (a and b), which can lead to a difference in their physicochemical properties. Thus, for tetra-*cis*-substituted MPcs, there is a high probability of manifestation of regioselectivity, which reduces the steric repulsion between adjacent bulky substituents (Figure 11a). In the case of tetra-*trans*-substituted MPcs, regioisomers are possible. In addition, the possibility of the formation of tetrasubstituted MPcs with different *cis*- and *trans*-orientations of 4-tritylphenoxy substituents within the same metal complex cannot be ruled out.

A series of copper (4-tritylphenoxy)phthalocyanines was synthesized in [4]. The authors note that the synthesized compounds exhibit mesomorphic properties and suggest that their mesomorphism originates from thermal fluctuations due to the free rotation of the bulky substituents. However, our data on barriers of internal rotations (Figure 3, Figure 4 and Figure 5) show that the assertion of the authors of [4] about the free rotation of phenoxy and trityl groups is untenable. Nevertheless, it seems that the copper(II) phenoxy(phthalocyaninato) complex with trans-oriented 4-tritylphenoxy substituents has no serious steric obstacles to the formation of a columnar mesophase (Figure 11b).

## 3. Methods and Materials

### 3.1. Synthesis

Synthesis of TPPN was carried out according to the scheme used in [7,13], which is closely repeated in [4] and is described in detail in Appendix A.

The product was characterized by mass spectrometry (LDI–TOF, negative mode), ^1^H NMR (CDCl_3_), ^13^C NMR (CDCl_3_), FTIR spectrum, and elemental analysis methods. The results are given in Appendix A.

### 3.2. The Infrared Spectrum

IR spectrum was recorded for the films deposited from a solution on the KRS-5 slides in the range of 400–4000 cm^−1^ using an Avatar 360 FTIR spectrometer. (Details in Section 2.2).

### 3.3. Synchronous Gas Electron Diffraction and Mass Spectrometric Experiment

The gas phase electron diffraction patterns and mass spectra were recorded simultaneously using the technique described in Refs. [14,15] at two nozzle-to-plate distances (338 and 598 mm). The conditions of the synchronous gas-phase electron diffraction/mass spectrometric (GED/MS) experiments are shown in Appendix A.

The electron wavelength was obtained by polycrystalline ZnO. The optical densities were measured by a computer-controlled modified MD-100 microdensitometer [16] with a step of 0.1 mm along the diagonal of the plate. A 10 × 130 mm region was scanned. The number of equidistant scan lines was 33. The total intensity curves were obtained in the ranges s = 1.3–16.7 Å^−1^ and s = 3.0–28.6 Å^−1^.

Simultaneously with the registration of electron diffraction patterns, the mass spectra of TPPN vapors were recorded. According to the mass spectrometric data, only TPPN is present in the vapor, no volatile impurities are observed. Table 4 shows that the mass spectra of electron ionization (U_ioniz_ = 50 V) contain signals of the parent ion [M]^+^ ([C_33_H_22_N_2_O]^+^) and lighter ions formed as a result of dissociative ionization of molecules: Ions formed upon breaking C_t_-C_t1_ bonds (see Figure 1 for atom numbering), with the loss of one [M-(C_6_H_5_)]^+^, two [M-(C_6_H_5_)_2_]^+^ and three [[M-(C_6_H_5_)_3_]^+^ phenyl groups of the trityl fragment, and additionally the loss of a substituent CN [M-(C_6_H_5_)_3_-CN]^+^, ions with loss of trityl fragment [M-C(C_6_H_5_)_3_]^+^ and group CN [M-C(C_6_H_5_)_3_-CN]^+^, trityl ion [C(C_6_H_5_)_3_]^+^, as well as the high intensive [C(C_6_H_5_)_2_]^+^ ion. In addition to ions associated with the detachment of individual groups, the mass spectrum contains ions related to the dissociative ionization of trityl phenyl fragments with the elimination of C_2_H_n_ and C_3_H_n_ groups ([M-C_3_H_3_-C_2_H_2_]^+^ and [(C_4_H_4_)_2_CC_3_H_3_]^+^. The most intense is the ion with stoichiometry [M-C_6_H_5_]^+^.

### 3.4. Computational Details

Quantum-chemical modeling of the TPPN molecule was implemented using the Gaussian 09 program [17]. The calculations were performed using the density functional theory (DFT) method, variants of hybrid functionals: B3LYP [18,19,20], PBE [21], CAM-B3LYP [22], as well as the B97D functional [23], which considers dispersion interactions. The atoms of carbon, oxygen, nitrogen, and hydrogen were described using correlation-consistent three-exponential valence bases cc-pVTZ [24].

Using the DFT/B3LYP/cc-pVTZ method, the potential energy surface (PES) was scanned along torsion coordinates, and the presence of four conformers was established. For each conformer of the TPPN molecule, the geometry optimization was performed (see Section 2.1) without symmetry restrictions. The calculation of the frequencies confirmed the correspondence of the optimized structures to the PES minima, and the calculated frequencies were used in the interpretation of the experimental IR spectrum.

## 4. Conclusions

The TPPN compound is thermally stable up to at least 200 °C, as evidenced by the data of the GED/MS experiment. This property of TPPN can be used in modeling CVD processes for obtaining thin films for various purposes.

The presence of bulky 4-tritylphenoxy substituents in the β-position can lead to regioselectivity of substituted phthalocyanines.

Using the DFT method with B3LYP, PBE, CAM-B3LYP, and B97D functionals, it was shown that the molecule TPPN has four conformers with the same structure of the -Ph-C(Ph)_3_ fragment and different positions of the -O-PhN group. All conformers have a significant dipole moment and similar energies of frontier orbitals. At the HOMO → LUMO transition, the electron density is transferred from the trityl to the phthalonitrile fragment.

The structure of transition states between conformers is determined. The close values of the total energy of the conformers and high transition barriers suggest that the conformational diversity of TPPN is formed during the synthesis of the compound.

All vibrational bands in the experimental IR spectrum of TPPN have been assigned. A slight difference was noted in the theoretical IR spectra of the conformers, which, however, made it possible to determine that TPPN consists of a mixture of conformers in the condensed state.

The elemental composition of the ions registered in the mass spectrum of electron ionization has been established in saturated vapor TPPN.

GED study of saturated vapor TPPN was performed. The experimental geometrical parameters of the conformers are found, which agree with the calculated ones within the experimental error. Joint analysis of GED and QC data indicates that at T = 478 K, the gas phase of TPPN consists of a mixture of conformers.

It can be assumed that the use of TPPN for the synthesis of tetrasubstituted metal phthalocyanines will lead to the formation of conformers that differ significantly in the spatial structure of peripheral fragments and retain the conformational features of TPPN.

Moreover, within the same complex, 4-tritylphenoxy substituents can be in different conformations. This feature of such complexes can be used for practical purposes.

## Data Availability

Not applicable.

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
