# Peer review of "Molecular Structure, Vibrational Spectrum and Conformational Properties of 4-(4-Tritylphenoxy)phthalonitrile-Precursor for Synthesis of Phthalocyanines with Bulky Substituent"

_ijms, 2022, doi:10.3390/ijms232213922_

Round 1

Reviewer 1 Report

This paper presents a very good work by Tverdova et al. on certain properties of 4-(4-tritylphenoxy)phthalonitrile (TPPN), which is a precursor for synthesis of very important metal phthalocyanines. The authors have studied conformations of TPPN in detail, and showed that the mixture of conformers is contained both in the condensed and in the gas phase, a feature that can be preserved in the metal complexes derived from TPPN.

There are two technical issues I would like to point out:

1. The authors should add a figure in the introduction section with structural fomula of TPPN.

2. The existing figures should be of better quality.

I recommend the acceptance of this paper after these minor revisions are done.

Author Response

Dear Reviewer,

We are grateful for your careful reading of our article and for constructive recommendations for its improvement. We agreed with your recommendations and made corrections to the manuscript.

Comment 1: The authors should add a figure in the introduction section with structural formula of TPPN.

Response: Scheme 1 with a structural formula of TPPN was added to Introduction section.

Comment 2: The existing figures should be of better quality.

Response: The quality of figures was increased. The zip file with all figures and schemes of high quality in pdf format was submitted together with revised manuscript.

Reviewer 2 Report

in the text authors say: "The synthesis of TPPN was performed according to the procedure described in [8]" this reference is dificult to access, so I think they should include the experimental procedure in the suppl. mat. I would also appreciate they comment why they use this procedure an not the one described in: J. Mater. Chem., 2012, 22, 14418 DOI: 10.1039/c2jm32284f where they say fo follow the method by "N. E. Galanin, L. A. Yakubov, E. V. Kudrik and G. P. Shaposhnikov, Russ. J. Gen. Chem., 2008, 78, 1436–1440."

I also missed the citation of Takagi's paper, since they study similar compounds.

A scheme with the preparation of phthalocyanine should be included, to have at least once the molecules in two dimensions would help to the reader.

Author Response

Dear Reviewer,

We are grateful for your careful reading of our article and for constructive recommendations for its improvement. We agreed with your recommendations and made corrections to the manuscript.

Comment 1: in the text authors say: "The synthesis of TPPN was performed according to the procedure described in [8]" this reference is difficult to access, so I think they should include the experimental procedure in the suppl. mat. I would also appreciate they comment why they use this procedure an not the one described in: J. Mater. Chem., 2012, 22, 14418 DOI: 10.1039/c2jm32284f where they say fo follow the method by "N. E. Galanin, L. A. Yakubov, E. V. Kudrik and G. P. Shaposhnikov, Russ. J. Gen. Chem., 2008, 78, 1436–1440."

Response: We described all stages of synthesis of TPPN in Supplementary materials.

Indeed, Ref. 8 (in revised manuscript it is Ref. 10) is difficult to access. Ref. 8 reports the overall scheme of synthesis without the methodological details. In contrary, article "N. E. Galanin, L. A. Yakubov, E. V. Kudrik and G. P. Shaposhnikov, Russ. J. Gen. Chem., 2008, 78, 1436–1440" reports the details of mentioned synthesis, which were reproduced by Takagi differing some features.

In revised manuscript we also followed the scheme of Ref. 8, however, some changes in methodic were made in comparison with Galanin and Takagi. In Supplementary materials to reviewed manuscript we describe used methodic of the synthesis of TPPN including the synthesis of precursors.  

Comment 2: I also missed the citation of Takagi's paper, since they study similar compounds.

Response: Thank you very much for the reminder. We forgot about this article. We included a mention of this article in the Introduction and discussed some of its results in the Discussion section.  (“A series of copper (4-tritylphenoxy)phthalocyanines was synthesized in [4]. The authors note that the synthesized compounds exhibit mesomorphic properties and suggest that their mesomorphism originates from thermal fluctuations due to free rotation of the bulky substituents. However, our data on barriers of internal rotations (Figures 3,4,5) show that the assertion of the authors of [4] about the free rotation of phenoxy and trityl groups is untenable. Nevertheless, it seems that the copper(II) phenoxy(phthalocyaninato) complex with trans-oriented 4-tritylphenoxy substituents has no serious steric obstacles to the formation of a columnar mesophase (Figure 11b)”.

Comment 3: A scheme with the preparation of phthalocyanine should be included, to have at least once the molecules in two dimensions would help to the reader.

Response:  A scheme with the preparation of TPPN and phthalocyanine was given in Supplementary materials. Section 1.

Round 2

Reviewer 2 Report

I am satisfied with authors corrections